# Genomic Characterization of Predominant Delta Variant (B.1.617.2 and AY.120 Sub-Lineages) SARS-CoV-2 Detected from AFI Patients in Ethiopia During 2021–2022

**DOI:** 10.3390/genes16111366

**Published:** 2025-11-11

**Authors:** Musse Tadesse Chekol, Dejenie Shiferaw Teklu, Adamu Tayachew, Wolde Shura, Admikew Agune, Aster Hailemariam, Aynalem Alemu, Mesfin Wossen, Abdulhafiz Hassen, Melaku Gonta, Neamin Tesfay, Tesfu Kasa, Nigatu Kebede

**Affiliations:** 1Public Health Emergency Management Center, Ethiopian Public Health Institute, Addis Ababa P.O. Box 1242, Ethiopia; 2Aklilu Lemma Institute of Pathobiology, Addis Ababa University, Addis Ababa P.O. Box 1242, Ethiopia

**Keywords:** SARS-CoV-2, mutation, delta variant, genetic characteristics, Ethiopia

## Abstract

**Background**: The Delta variant of SARS-CoV-2 virus, one of the alarming variants of concern (VOC) with a distinct mutation characteristic, was immensely detrimental and a significant cause of the prolonged pandemic waves. This study aimed to analyze the genetic characteristics of the predominant Delta variant in acute febrile illness (AFI) patients in Ethiopia. **Method**: Nasopharyngeal swab samples were collected from AFI patients in four hospitals from February 2021 to June 2022 and tested for SARS-CoV-2 by using RT-qPCR. Of 101 positive samples, 48 stored specimens were re-tested, and 26 with sufficient RNA quality (Ct < 30) were sequenced using whole-genome sequencing to identify variants of concern, specific virus lineages and mutation features. **Result**: Delta variants (21J clade) were found predominant among all the sequenced SARS-CoV-2 isolate (80.8%, 21/26). AY.120 (46.2%) and B.1.617.2 (26.9%) were the predominant sub-lineages of the Delta variant. Omicron (21k, Pango BA.1.1/BA.1.17/BA.1) and Alpha (20I, Pango B.1.1.7) variants accounted for 11.5% and 7.7% of the total sequenced samples. Phylogenetic analysis showed evidence of local transmission and possible multiple introductions of SARS-CoV-2 VOCs in Ethiopia. The number of mutations increases dramatically from Alpha (~35 avg) to Delta (~42 avg) to Omicron (~56 avg). The Delta variant revealed a spike mutation on L452R and T478K and P681R, and was characterized by the double deletion E156-F157- in Spike protein. **Conclusions**: The findings are indicative of a gradual change in the genetic coding of the virus underscoring the importance of ongoing genomic surveillance to track the evolution and spread of SARS-CoV-2 and other emerging virus.

## 1. Introduction

SARS-CoV-2, an RNA virus, has a high mutation rate due to the error-prone nature of viral RNA-dependent RNA polymerase (RdRp) and recombination events, allowing for rapid evolution and adaptation [1,2,3]. SARS-CoV-2’s ongoing molecular evolution has enhanced its global spread, resulting in a series of Variants of Concern (VOCs) that have influenced the pandemic’s course [1,4,5]. These VOCs have unique mutations in the spike (S) protein and other genomic regions [6,7,8,9,10,11]. Enhanced binding affinity to the human angiotensin-converting enzyme 2 (ACE2) receptor, greater transmissibility, and immune evasion have all been linked to mutations in the receptor-binding domain (RBD) of the S protein [3,12]. Furthermore, genomic insertions and deletions have been noted, which have been linked to antigenic drift and changed viral fitness [13,14].

This virus is projected to evolve at a moderate rate (∼10^−3^–10^−4^ substitutions/site/year) which has varied throughout time. Recombination events have been comparatively rare; however, they have been found in the spike (S) protein, which contributes to variations in the viral variants [15,16]. Positive selection has been the key force of adaptation, especially on the spike protein, where mutations in the receptor-binding domain (RBD), such as N501Y, E484K, and L452R, have improved transmissibility and allowed immune evasion from both natural and vaccine-induced immunity [6,7,8,9,10,11]. Alpha (B.1.1.7), Beta (B.1.351), Gamma (P.1), Delta (B.1.617.2), and Omicron (B.1.1.529) are some of the variants of concern (VOCs) that have emerged since SARS-CoV-2 was first discovered in Wuhan, China [1,4,5]. The emergence of these VOCs, defined by their unique constellations of mutations, exemplifies this adaptive evolution, resulting in waves of infection with altered epidemiological characteristics [12,13,14]. These global evolutionary mechanisms provide the essential framework for understanding the virus’s behavior on a regional scale, including in Ethiopia.

Due to its increased transmissibility and capacity for immune evasion, the SARS-CoV-2 Delta variant (B.1.617.2), which was initially discovered in India in late 2020, swiftly rose to prominence as a dominant strain worldwide [1,4,14]. The Delta variant was responsible for more than 90% of sequenced cases in multiple countries by the middle of 2021, causing major COVID-19 outbreaks that put a pressure on healthcare systems around the globe [1,4,5,14,15].

In Ethiopia, genomic surveillance has confirmed that the pandemic’s waves were driven by the sequential importation and community transmission of commonly known circulating VOCs. Studies have documented the introduction and dominance of variants including Alpha (B.1.1.7), Beta (B.1.351), Delta (B.1.617.2), and Omicron (B.1.1.529) and its sub-lineages in the country [17,18]. While the circulation of these defined variants mirrors global trends, preliminary data also points to the possible emergence of unique local mutations, particularly within the Omicron wave, suggesting potential regional evolutionary pressures [17,18]. However, the extent of these locally emerging mutations and their phenotypic consequences remain inadequately explored due to limited and intermittent genomic sequencing.

SARS-CoV-2 genetic characterization has been crucial for understanding the mutation characteristics, tracking viral transmission, determining toxicity, and influencing public health actions such as vaccine development [16]. There are limitations in publications on the genetic evolution and mutation patterns of SARS-CoV-2 variants in Ethiopia, despite international genomic surveillance efforts, emphasizing the necessity of localized research to guide public health actions [17,18]. This study intends to analyze the genetic characteristics of the predominant Delta variant in Ethiopia, aiming to address the existing knowledge gap and contribute to advancing worldwide comprehension of SARS-CoV-2 development in low- and middle-income environments.

## 2. Methods and Material

### 2.1. Ethical Clearance

Ethical clearance was obtained from the Ethiopian Public Health Institute (EPHI-IRB-254-2020) and Addis Ababa University (IRB/19/025). All participants provided written informed consent prior to enrollment.

### 2.2. Study Population and Specimen Collection

Genetic analysis of SARS-CoV-2 was conducted using repository nasopharyngeal samples collected from AFI patients in a cross-sectional study during February 2021 to June 2022 in four hospitals (St. Paul Hospital, Jimma University Hospital, Hiwot Fana Hospital, and Gonder University Hospital) [19]. The nasopharyngeal swabs were collected from enrolled cases using a sterile COPAN brand universal transport medium containing 1–3 mL Viral Transport Media. The collected sample was vortexed and aliquoted in two cryovials for molecular testing at the National Influenza Center (NIC), Ethiopian Public Health Institute (EPHI). The samples were stored at 4 °C until transported by trained postal service officers to the NIC at EPHI. EPHI’s NIC (laboratory) identified respiratory viruses from nasopharyngeal swabs.

A total of 737 samples were collected, of which 101 (13.7%) tested positive for SARS-CoV-2 by real-time PCR. From these, 48 stored positive samples were randomly selected and re-tested for SARS-CoV-2, and 26 samples with cycle threshold (Ct) values < 30 were prepared for the Whole-genome sequencing. The enrolled cases were inpatients and outpatients presented at the selected facilities, and who met the case definition criteria for AFI with a predefined inclusion and exclusion criteria.

### 2.3. Viral Load Determination and cDNA Synthesis

Nucleic acid was extracted from the swabs using the MagBioplus Virus RNA Purification Kit II by MGISP-NE32 automated extractor. Real-time PCR was conducted on an ABI 7500FAST system (Life Technologies, Carlsbad, CA, USA) using primers provided by CDC International Reagent Resource (CDC-IRR), a biological reagent repository established to provide better access to laboratory reagents. Samples with Ct values of 30 or less were deemed uniformly suggestive of a substantial viral load for the purposes of genome sequencing, eligible for whole-genome sequencing, and suitable for downstream analysis, Figure 1.

### 2.4. Whole-Genome Sequencing and Bioinformatics Analysis

Whole-genome sequencing was performed on SARS-CoV-2 positive samples with a cycle threshold (Ct) value of less than 30. Viral RNA was converted to cDNA using the LunaScript RT SuperMix (New England Biolabs, Ipswich, MA, USA). Subsequently, the SARS-CoV-2 genome was amplified using the Midnight RT PCR Expansion Kit (EXP-MRT001, Oxford Nanopore Technologies, Oxford, UK), which generates approximately 1.2 kb overlapping amplicons. Library preparation was carried out using the Rapid Barcoding Kit 96 (SQK-RBK110.96, Oxford Nanopore Technologies), and sequencing was performed on the MiniON platform (Oxford Nanopore Technologies) using R9.4.1 flow cells (FLO-MIN106, Oxford Nanopore Technologies). Real-time base-calling and demultiplexing were executed within the MinKNOW software (v. 23.04.13, Oxford Nanopore Technologies).

Raw FAST5 reads were base-called and demultiplexed using Guppy (v. 6.4.6, Oxford Nanopore Technologies) within the MinKNOW interface. The resulting FASTQ files were processed using the wf-artic workflow within the Nextflow-based nf-core/viral-recon pipeline (v.2.6.0), which is optimized for amplicon-based SARS-CoV-2 data. This pipeline performed read trimming, alignment to the SARS-CoV-2 reference genome Wuhan-Hu-1 (NC_045512.2) using minimap2 (v.2.24), and consensus genome generation using ivar (v.1.3.1). Quality control thresholds for the consensus genomes included a minimum depth of coverage of 30×, a genome coverage of >95%, and fewer than 3000 ambiguous bases (N’s). The mean read length after trimming was approximately 1200 bp, consistent with the expected amplicon size. SARS-CoV-2 raw reads obtained from Oxford Nanopore sequencing machine were assembled by using Genome Detective Platform (Version 2.15.1, Abiotic, Tiel, Netherlands) (https://www.genomedetective.com/; accessed on 23 August 2024) to generate consensus genomes.

### 2.5. Variant Calling and Phylogenetic Analysis

Variant calling and lineage assignment were conducted using two primary tools. The consensus sequences were analyzed with Nextclade CLI (v.3.8.2) for quality control, clade assignment, and identification of amino acid substitutions and deletions. PANGOLIN (v.4.3, accessed via the web interface at pangolin.cog-uk.io) was used for dynamic Pango lineage classification. For phylogenetic analysis, a maximum-likelihood tree was inferred using the Augur pipeline (v.23.1), which employs IQ-TREE (v.2.2.0) with the General Time Reversible (GTR) model of nucleotide substitution. The resulting tree was visualized using Auspice.us. The complete genome sequences generated in this study have been deposited in the GISAID database and are publicly available.

### 2.6. Statistical Analysis and Visualization

Python scripts (v3.8) were created to process the CSV output obtained from the NextClade pipeline. Descriptive statistics were used to summarize and highlight the main features of the dataset. The findings were presented through graphs and tables. For the analysis of the mutational profile, a subset of amino acid substitutions and deletion with frequencies above 75% were examined.

## 3. Results

### 3.1. Socio-Demographic and Clinical Characteristics

The randomly selected SARS-CoV-2 positive samples for genetic analysis (*n* = 48 participants consented) were composed of patients with a median age of 43 years old (range: 19–90 years old), where 54% were male and 46% were females (Table 1, below). Sudden onset of fever (>38 °C measured axillary body temperature) was observed in all the participants; Cough (83%), Sore Throat (65%), Headache (71%) and Muscle ache (63%) were the most commonly reported symptoms among participants. Most of participants (90%) in this study were inpatients admitted with a complaint of acute febrile illness.

### 3.2. Assembly and Sequencing Metrics of SARS-CoV-2 Raw Read Sequences

The Genome Detective platform, which generated the consensus sequence of 26 isolates and provided metrics like the number of reads, depth of coverage, nucleotide identity, amino acid identity, and genome coverage. The study employed rigorous and well-documented quality control measures. An average depth of coverage ranging from 279 to 1014, with excellent genome coverage (>95%), and high nucleotide identity (>99.8%) collectively confirm that the genomic data generated was of high quality and that the findings on mutations and variants are reliable (Table 2).

### 3.3. SARS-CoV-2 Genetic Diversity over the Study Period

In the study period the analysis of genetic diversity showed the predominance of Delta variants or 21J clade (80.8%, 21/26) among all the sequenced SARS-CoV-2 isolate according to WHO and the Next strain naming system. The Delta variant was also sub-classified as AY.120 (*n* = 12, 46.2%) and B.1.617.2 (*n* = 7, 26.9%) under the Pangolin Lineage naming systems. Omicron and Alpha (Pango B.1.1.7) variants accounted for 11.5% (*n* = 3) and 7.7% (*n* = 2) of the total sequenced samples (Table 3).

### 3.4. Patterns of SARS-CoV-2 Genetic Variation During the Study

The analysis of the temporal trends in the prevalence of the three SARS-CoV-2 variants during the study period showed that there were Alpha variants in May 2021 with low prevalence. The Delta variant emerged as the predominant variant starting from July 2021 and remained dominant until November 2021. The Omicron variants were detected for the first time in November 2021 and observed again in December 2021 and February 2022, see Figure 2.

### 3.5. Nucleotide and Amino Acid Mutation Analysis in SARS-CoV-2 Genes

The mutational analysis reveals a clear evolutionary progression from Alpha to Omicron, characterized by a significant increase in genetic changes, particularly in the Spike protein, which enhances the virus’s ability to evade immunity and infect cells. The number of mutations increases from Alpha (~35 avg) to Delta (~42 avg) to Omicron (~56 avg) which reflects the average number of mutations across the entire genome for each variant, relative to the original Wuhan-Hu-1 reference strain (NC_045512.2). The Alpha variant featured a spike mutation on N501Y (improved cell entry) and P681H (enhanced infectivity). The Delta variant revealed a spike mutation on L452R and T478K (strong immune evasion) and P681R (high infectivity), and was characterized by the double deletion E156-F157- in Spike. The Omicron variant contains over 30 changes in Spike alone. Key mutations include G339D, S371L, N440K, S477N, T478K, E484A, Q498R, N501Y (all in the receptor-binding domain, allowing it to evade most existing antibodies), Table 4 and Table 5.

### 3.6. Phylogenetic Analysis

The phylogenetic tree depicted in Figure 2 below shows three major clusters (clades) of SARS-CoV-2 variants based on the known variant classification system and distinct branching patterns. More genetically and closely related consensus sequences are grouped into the same cluster and each cluster forms a distinct branch or set of branches on the tree. On this figure at the bottom left, there is a reference genome, which is the original SARS-CoV-2 isolated in late 2019. All the current SARS-CoV-2 variants are descended from this initial strain. The first cluster, cluster 1, was the smallest (*n* = 2), comprised Alpha variants. The second cluster consists of Omicron (*n* = 3), which is composed of entirely Omicron variants. The third cluster, occupying the upper part of the tree, comprised the Delta variant (*n* = 21) that has many sub-lineages within it (Figure 3).

When we see the common ancestral relationships of the variants in the three clusters, the phylogenetic tree shows that Cluster 1 (Alpha variant) is the older group, branching off earlier from the common ancestor of all sequences (i.e., from the Wuhan). The Delta variant shows several sub-branches, indicating ongoing evolution and diversification within this variant. The significant genetic divergence of Omicron, as evidenced by its long branch length, is consistent with its highly mutated genome (Figure 3).

## 4. Discussion

In this study, SARS-CoV-2 genomes were characterized using nasopharyngeal specimens collected from patients with AFI between February 2021 and June 2022 in four hospitals in Ethiopia. It was found that the Delta variants (lineage B.1.617.2 and AY.120) were predominant among specimens with high viral load, accompanied by characteristic spike and non-spike protein mutations; a smaller proportion of Alpha and Omicron variants were also detected. The high depth and coverage of sequencing enabled reliable variant calling and phylogenetic placement, allowing us to observe diversification of Delta, including the emergence of local sub-lineage signatures.

The sequencing metrics found in this study show high-quality SARS-CoV-2 genomic data, with consistently high nucleotide and amino acid identities across all isolates. The nucleotide identity averaged 99.8%, whereas the amino acid identity was 99.7%, showing little sequence divergence from the reference genome. These findings are consistent with recent investigations from East Africa, which have revealed similar high sequence identities ranging from 99.68% to 99.92% and amino acid identities of around 99.94% [17,18,20].

The depth of coverage in this dataset ranging from 279 to 1014 ensured a reliable variant calling and sequence assembly. This degree of coverage supports the accuracy of consensus sequences and mutation detection and is comparable to earlier research conducted in Africa, where sequencing depths of more than 500× have been documented for SARS-CoV-2 genomes. The degree of coverage is similarly comparable to other studies conducted in Northeast Ohio (average coverage: 1081×; range: 223–2235×) and France (range: 279–1014×) [21,22]. Furthermore, the average genome coverage was 99.4%, with very slight differences amongst isolates. It is possible that the somewhat decreased genome coverage of some isolates (96.0% and 95.7%, for example) was caused by issues with RNA quality, sequencing depth, or the difficulty of sequencing particular genomic areas.

The genomic data from this study provide a molecular snapshot of the COVID-19 pandemic in Ethiopia, confirming that the Delta variant was the primary driver of the country’s third wave from July to October 2021. The overwhelming predominance of the Delta variant (21/26, 80.8%), particularly the AY.120 sub-lineage, suggests that this sub-lineage may have had a selective advantage for local transmission, leading to a significant community outbreak during this period. Perhaps, this may not represent the full genetic diversity of SARS-CoV-2 in Ethiopia throughout the entire epidemic. This result was consistent with the GISAID data analysis (up to July 2021), which showed that the Delta variant’s prevalence ranged from 67.6% to 98.3% of all circulating variations in Africa [21]. Before Omicron surpassed it in early 2022, Delta first appeared in the spring of 2021, gained dominance in the middle of the year, and continued to be significantly higher until the end of 2021 [21,23]. Furthermore, Africa’s third COVID wave was caused by Delta, which also accounted for more than one-third of all infections on the continent [21,24]. Compared to earlier variants, Delta has a little increase in immune evasion and is more transmissible, which leads to its increased prevalence [6,25].

The most distinctive finding in this study was the predominance of the Delta sub-lineage AY.120, which accounted for 46.2% (*n* = 22) of all sequenced samples. This suggests that AY.120 was not just a sporadic import but likely underwent significant local transmission within Ethiopia, potentially becoming a locally adapted strain during the study period. Regionally, Ethiopia’s experience aligns with broader African trends where Delta dominated mid-2021. However, the specific dominance of the AY.120 sub-lineage in Ethiopia differs from genomic reports in neighboring countries like Kenya and Uganda, which reported a more diverse mix of Delta sub-lineages (e.g., AY.36, AY.46) [20,24]. This unique sub-lineage profile suggests that Ethiopia’s epidemic was shaped by specific international introduction events, potentially from Europe or Asia where AY.120 was also detected, followed by localized transmission rather than cross-border spread from immediate neighbors. This underscores the importance of local genomic surveillance to track distinct transmission dynamics that global or regional data may not capture.

According to a comparative study carried out in England, the increase in AY.120 detections suggested an estimated rate of spatial growth of 6.56 (6.16–6.97), and there were 18,400 detections during the same study period, supporting the predominance observation of sub-lineage AY.120 in this study [26]. This indicates that during the middle of 2021, a Delta infection wave, evolving both locally and internationally, included the sub-lineages AY.120 and B.1.617.2. In 74 nations, including the UK (85%), India (3%), USA (3%), Germany (2%), and France (1%), AY.120 sub-lineage detections were reported, with a global prevalence of 1% [27]. Furthermore, AY.120 sub-lineage was reported in 74 countries including in UK (85%), India (3%), USA (3%), Germany (2%), and France (1%) with a 1% global prevalence [27].

Although a large number of AY.120 lineages have been identified in multiple countries, relatively little information has been published regarding the epidemiological and genetic characteristics of these lineages. Crucially, there is no indication of a protective contextual effect, such as rising vaccination rates or natural immunity, that would explain the selective advantage of this sub-lineage or biological selection pressure favoring the growth advantage of emerging lineages [26]. While the study does not perform a direct statistical correlation, the timing of the Delta wave coincided with a period of low vaccination coverage in Ethiopia, which likely facilitated the variant’s rapid spread. The high proportion of inpatients (90%) among sequenced cases is consistent with global reports of increased clinical severity associated with the Delta variant, including a higher risk of hospitalization compared to previous variants.

However, the core Delta mutations in the spike protein (L452R, T478K, P681R, and others) that are known to increase transmissibility and decrease vaccine susceptibility may be shared by AY.120 because it is a member of the larger Delta clade [28]. No discernible variations in clinical severity were reported in studies on comparable AY lineages (e.g., AY.28, AY.104, AY.3, AY.122); however, AY.120 is not included in these data [29]. Given the results that have been reported, it would appear that this merits more research.

Delta variant sub-lineage B.1.617.2, the other prominent sub-lineage found in this study, was initially discovered in Maharashtra, India, in October 2020, and was designated as a Variant of Concern (VOC) by the WHO on 11 May 2021 [1,4,5,14,30,31]. Because of its high transmissibility and shorter transmission cycles, the Delta variant showed a remarkable global growth. By 15 June 2021, the U.S. CDC reported that B.1.617.2 had been found in at least 66 countries and had spread to over 130 countries [1,5,14,25]. According to several studies, it is 40–60% more transmissible than Alpha; R_0_ is roughly 5 compared to 2.7 for wild-type [5,14,31].

Numerous distinguishing mutations, especially in the spike (S) protein, which increase transmissibility and impact immunological interactions, are present in Delta variant sub-lineage B.1.617.2. Similarly to the finding in this study, few other studies reported that the most prevalent key spike mutations are T19R, G142D, Δ157–158, R158G, L452R, T478K, D614G, P681R, and D950N [32]. The mutation profile of the circulating Delta variants explains their success. The critical spike mutations L452R and T478K are known to enhance binding to the human ACE2 receptor and confer significant immune evasion from neutralizing antibodies. The P681R mutation facilitates more efficient viral entry into cells by enhancing spike protein cleavage. Furthermore, the E156-F157 deletion is thought to contribute to antigenic drift by removing an epitope, thereby weakening the antibody response from prior infection or vaccination [11,33,34]. Other mutations that may affect viral replication, assembly, and immunological interactions include D950N in S2, as well as modifications in ORF1a/b, M, N, ORF3a, ORF7a/b, and ORF8 [30].

The Delta variant’s L452R and T478K spike mutations worked in concert to prevent neutralizing antibody binding, providing considerable immune evasion against immunity caused by infection and vaccination. At the same time, the P681R mutation significantly improved the effectiveness of spike protein cleavage, which is necessary for viral entry, resulting in significantly higher viral loads and infectivity. The E156-F157 deletion, which is thought to form a glycan shield and add another layer of antibody avoidance, was added to these alterations. The rapid global dominance of the Delta variant was supported by this potent combination of increased transmissibility and robust immune evasion, which were driven by these precise molecular changes [35].

The Delta variants detected (including AY.120 and B.1.617.2) carried the characteristic suite of spike protein mutations such as L452R, T478K, P681R, and the double deletion E156-F157-, which are linked to increased transmissibility and immune evasion. While these are hallmarks of the Delta variant globally, their consistent presence in the Ethiopian clusters confirms the circulation of highly infectious strains and provides a baseline of the specific mutations circulating in the country.

Compared to earlier variants, the Delta variant is linked to more severe illness outcomes as reported by some studies. Data from Public Health England & Scotland indicated that the hospitalization was 2× riskier than Alpha. Ct values are lower (less than 30 more frequently), indicating higher infectivity; the viral load is approximately 1000 times greater than that of ancestral strains [30]. Perhaps there is lack of data indicating the severity of this variant in Ethiopia, studies conducted in other countries have indicated that Delta infection increases the probability of hospitalization by 120%, ICU admission by 287%, and fatality by 137% when compared to non-VOC strains.

The phylogenetic analysis indicates that the Delta variants in Ethiopia did not originate from a single introduction event. Instead, the data show clustering patterns suggestive of multiple independent introductions from international sources, followed by established local transmission chains. This underscores the role of global travel in virus importation and the subsequent community spread within Ethiopia. Similarly, studies in Zambia [36] that used phylogenetic analysis revealed indications of local transmission and potential numerous introductions of SARS-CoV-2 variants from various European and African countries corroborate this observation. These results highlight the interdependence of SARS-CoV-2 transmission networks as well as the part that worldwide travel and trade play in the virus’s propagation [36,37].

### Limitations of the Study

This study presented the genetic features of the Delta variant detected from AFI patients in Ethiopia where scientific data on genetic characteristics were scarce. The AY.120 strain of the Delta variant presented in this study were not thoroughly studied globally and very few publications were available; the findings on this study may contribute on providing the genetic features of this variant for the global database. Also, it might give an insight on mutation features of this Delta variant in the low- and middle-income countries (LMICs). This study is limited by a small number of sequenced isolates (*n* = 26). Nevertheless, the findings provide valuable insight on genetic information of the circulating SARS-CoV-2 like other published studies with few numbers of samples such as in Guinea (*n* = 19), Nigeria (*n* = 34), and Zhengzhou, China (*n* = 5); [38,39,40]. An association between the epidemiological factors and the identified variants was not investigated due to the limited sample size.

## 5. Conclusions

Delta variant sub-lineage AY.120 and B.1.617.2 were the predominant strain of SARS-CoV-2 virus circulating in Ethiopia during the study period. The mutational analysis reveals a clear evolutionary progression, characterized by a significant increase in genetic changes, particularly in the Spike protein. These findings provide partial but valuable insights into the genetic diversity of SARS-CoV-2 in Ethiopia and highlight the importance of continuous genomic surveillance and expanded sequencing efforts to track the spread and evolution of emerging variants.

## Figures and Tables

**Figure 1 genes-16-01366-f001:**
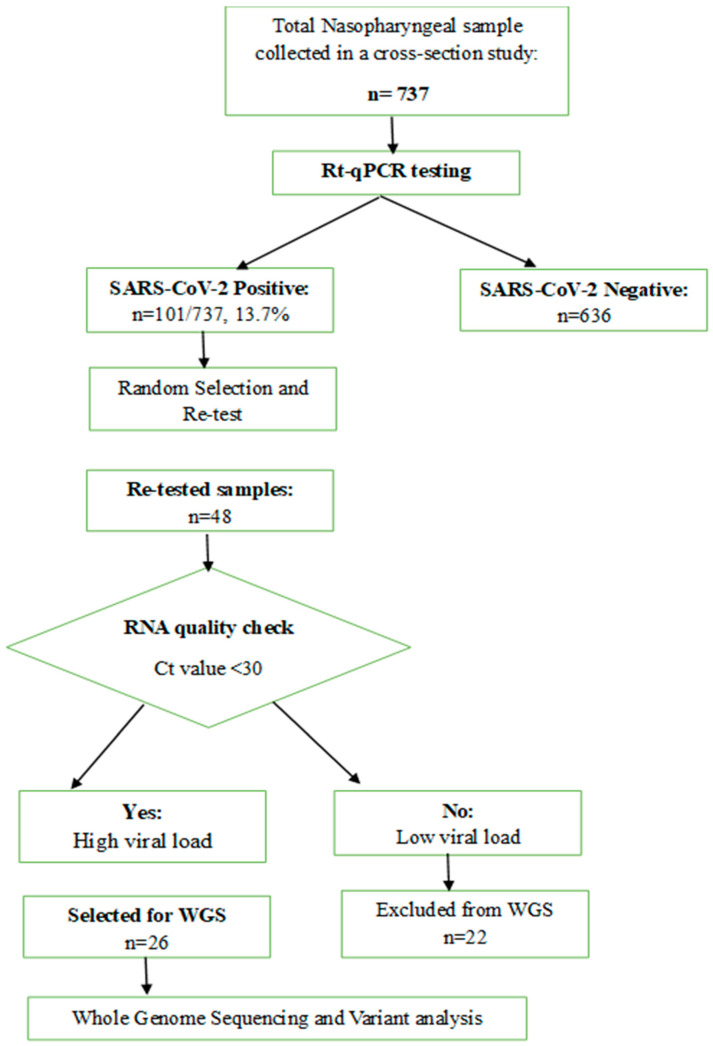
Flow Diagram of Sample Processing to Genomic Analysis.

**Figure 2 genes-16-01366-f002:**
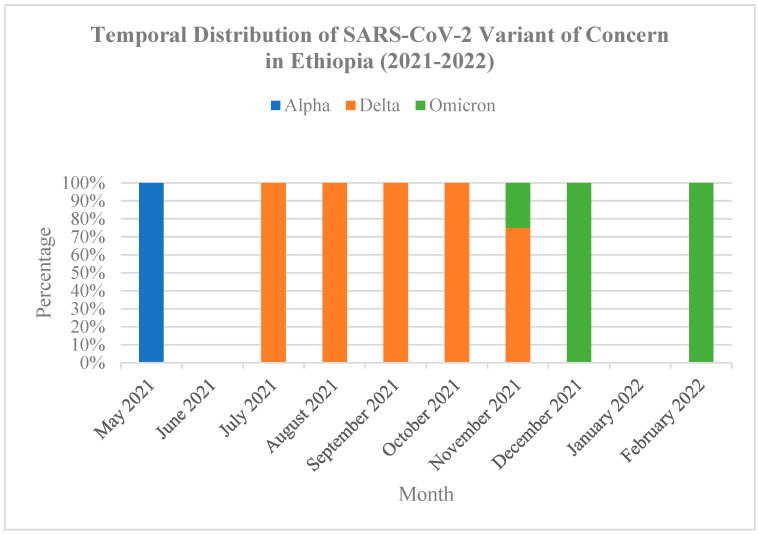
Temporal Distribution of SARS-CoV-2 WHO Variant of Concern (Alpha, Delta, and Omicron) by month in Ethiopia (2021–2022).

**Figure 3 genes-16-01366-f003:**
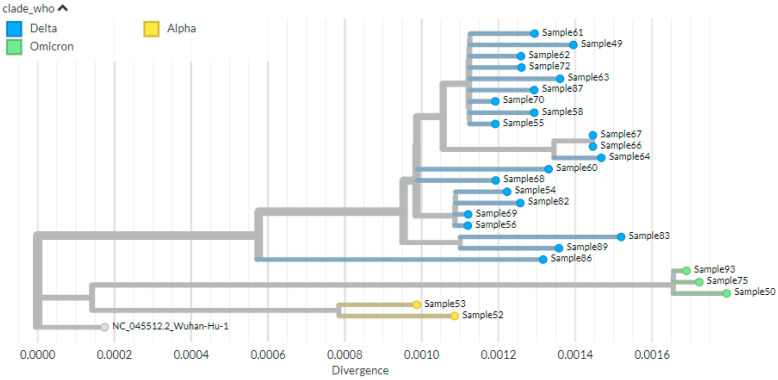
Phylogenetic tree visualized using Auspice.us and showing the evolutionary relationships of SARS-CoV-2 variant of concern. The tree is rooted on the reference genome NC_045512.2 (Wuhan-Hu-1). Major clades are labeled using the World Health Organization (WHO) nomenclature system (clade_who), which designates Variants of Concern (VOCs) Alpha, Delta, and Omicron. The horizontal axis (Divergence) indicates the genetic distance from the root, measured in nucleotide substitutions per site. Sample names are shown for sequences within the defined clades.

**Table 1 genes-16-01366-t001:** Socio-demographic and clinical characteristics of the study population (*n* = 48 participants).

Characteristics		Frequency (*n* = 48)	Percentage
Sex	Female	22	46%
	Male	26	54%
Age	15–24 Years old	3	6%
	25–44 Years old	22	46%
	45–64 years old	8	17%
	65+	15	31%
Symptoms	Fever	48	100%
	Cough	40	83%
	Sore Throat	31	65%
	Difficulty of breathing	27	56%
	Headache	34	71%
	Muscle ache	30	63%
	Arthralgia	16	33%
Admission Status	Inpatient	43	90%
	Outpatient	5	10%

**Table 2 genes-16-01366-t002:** Sequencing depth and accuracy for SARS-CoV-2 sequence according to Genome Detective Platform.

Barcode	# Reads	Depth of Coverage	Nucleotide Identity (%)	Amino Acid Identity (%)	Genome Coverage (%)
49	27,604	485	99.8	99.7	99.4
50	27,620	502	99.8	99.6	96.0
52	55,585	1014	99.9	99.9	99.4
53	15,574	279	99.9	99.8	99.4
54	28,642	486	99.9	99.7	99.4
55	42,675	769	99.9	99.7	99.4
56	51,054	915	99.9	99.7	99.4
58	27,377	485	99.9	99.7	99.4
60	51,101	931	99.9	99.7	97.7
61	43,354	771	99.9	99.7	99.4
62	27,608	495	99.9	99.7	99.4
63	55,379	979	99.8	99.7	99.4
64	53,559	959	99.8	99.7	99.4
66	26,429	466	99.8	99.7	99.2
67	53,016	937	99.8	99.7	99.4
68	37,235	666	99.9	99.7	98.2
69	26,743	467	99.9	99.7	97.5
70	33,827	595	99.9	99.7	99.4
72	35,043	610	99.9	99.7	99.2
75	37,569	646	99.8	99.6	99.4
82	46,077	841	99.9	99.7	99.4
83	17,440	293	99.8	99.7	95.7
86	23,716	430	99.9	99.7	99.4
87	37,558	662	99.9	99.7	99.4
89	21,772	360	99.8	99.7	99.4
93	54,886	949	99.8	99.7	99.4

**Table 3 genes-16-01366-t003:** The frequency and percentage of SARS-CoV-2 variants in three naming systems (WHO, Pangolin and Next strain).

WHO Variant N (%)		Pangolin Lineage N (%)		Next Strain Clade N (%)	
Alpha	2 (7.7)	B.1.1.7	2 (7.7)	20I	2 (7.7)
Delta	21 (80.8)	AY.120	12 (46.2)	21J	21 (80.8)
		B.1.617.2	7 (26.9)		
		AY.32	2 (7.7)		
Omicron	3 (11.5)	BA.1.1	1 (3.8)	21K	3 (11.5)
		BA.1.17	1 (3.8)		
		BA.1	1 (3.8)		

**Table 4 genes-16-01366-t004:** Mutation Characteristics of SARS-CoV-2 variants by sub-lineages.

Variant (Sub-Lineage)	Protein	High-Frequency Substitutions (>75%)	High-Frequency Deletions (>75%)
**Alpha (B.1.1.7)**	ORF1a	T1001I, A1708D, P2287S	S3675-, G3676-, F3677-
	ORF1b	P314L, G662S	
	ORF8	Q27, *R52I*, *K68*, Y73C	
	N	D3L, R203K, G204R, S235F	
	S	N501Y, D614G, P681H	H69-, V70-, Y144-
**Delta (B.1.617.2)**	ORF1a	A1306S, P2046L, P2287S, V2930L, T3255I, T3646A	
	ORF1b	P314L, G662S, P1000L, A1918V	
	ORF3a	S26L	
	ORF7a	V82A, T120I	
	ORF7b	T40I	
	ORF9b	T60A	
	M	I82T	
	N	D63G, R203M, G215C, D377Y	
	S	T19R, G142D, R158G, L452R, T478K, D614G, P681R, D950N	E156-, F157-
**Delta (AY.32)**	ORF1a	A1306S, P2046L, P2287S, V2930L, T3255I, T3646A	
	ORF1b	P314L, G662S, P1000L, A1918V, T2376I, R2613C	
	ORF3a	S26L	
	ORF7a	V82A, T120I	
	ORF7b	T40I	
	ORF9b	T60A	
	M	I82T	
	N	D63G, R203M, G215C, D377Y	
	S	T19R, G142D, R158G, L452R, T478K, D614G, P681R, D950N	E156-, F157-
**Delta (AY.120)**	ORF1a	V28I, A1306S, P2046L, S2048F, P2287S, V2930L, T3255I, T3646A	
	ORF1b	P314L, G662S, P1000L, A1918V, A2306T	
	ORF3a	S26L	
	ORF7a	V82A, T120I	
	ORF7b	T40I	
	ORF9b	T60A	
	M	I82T	
	N	D63G, R203M, G215C, D377Y	
	S	T19R, T95I, G142D, R158G, L452R, T478K, D614G, P681R, D950N	E156-, F157-
**Omicron (BA.1; BA.1.1, BA.1.17)**	ORF1a	K856R, A2710T, T3255I, P3395H, I3758V	L3674-, S3675-, G3676-
	ORF1b	P314L, I1566V	
	ORF9b	P10S	E27-, N28-, A29-
	M	D3G, Q19E, A63T	
	N	P13L, R203K, G204R	E31-, R32-, S33-
	S	A67V, T95I, G142D, Y145D, L212I, G339D, S371L, S373P, S375F, K417N, N440K, G446S, S477N, T478K, E484A, Q493R, G496S, Q498R, N501Y, Y505H, T547K, D614G, H655Y, N679K, P681H, N764K, D796Y, N856K, Q954H, N969K, L981F	H69-, V70-, G142-, V143-, Y144-, N211-

**Table 5 genes-16-01366-t005:** Summary of SARS-CoV-2 Variants, sub-lineages, and characteristic mutations detected in Ethiopia (2021–2022).

WHO Variant (Clade)	Pango Lineage (Sub-Lineage)	Frequency *n* (%) [N = 26]	Characteristic High-Frequency Spike Protein Mutations (>75%)	Characteristic Deletions (>75%)
Alpha (20I)	B.1.1.7	2 (7.7%)	N501Y, D614G, P681H	H69-, V70-, Y144-
Delta (21J)	AY.120	12 (46.2%)	T19R, G142D, R158G, L452R, T478K, D614G, P681R, D950N	E156-, F157-
	B.1.617.2	7 (26.9%)	T19R, G142D, R158G, L452R, T478K, D614G, P681R, D950N	E156-, F157-
	AY.32	2 (7.7%)	T19R, G142D, R158G, L452R, T478K, D614G, P681R, D950N	E156-, F157-
Omicron (21K)	BA.1.1, BA.1.17, BA.1	3 (11.5%), One (3.8%) for each sub linage	A67V, T95I, G142D, Y145D, G339D, S371L, S373P, S375F, K417N, N440K, S477N, T478K, E484A, Q493R, Q498R, N501Y, Y505H, D614G, H655Y, N679K, P681H, N764K, D796Y, N856K, Q954H, N969K, L981F	E27-, N28-, A29-, H69-, V70-, G142-, V143-, Y144-, N211-

## Data Availability

The raw data supporting the conclusions of this article will be made available by the authors, without undue reservation.

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
