# Peer review of "Genomic Characterization of Predominant Delta Variant (B.1.617.2 and AY.120 Sub-Lineages) SARS-CoV-2 Detected from AFI Patients in Ethiopia During 2021–2022"

_genes, 2025, doi:10.3390/genes16111366_

Round 1

Reviewer 1 Report

Comments and Suggestions for Authors

The manuscript presents genomic surveillance data of SARS-CoV-2 variants collected from Acute Febrile Illness (AFI) patients in Ethiopia between February 2021 and June 2022, focusing on the predominance and genomic characteristics of the Delta (B.1.617.2 and AY.120) lineages. The study is relevant and timely, as genomic data from sub-Saharan Africa remain limited, and understanding lineage diversity is important for pandemic preparedness and vaccine-response monitoring.

The paper is generally well organized, but it needs major revisions to improve scientific clarity, data presentation, and interpretation. The methodology, results, and discussion sections require additional detail to meet Genes’ expected rigor. Some phrasing and grammar also need refinement for scientific English.

1. Novelty and Scientific Value

  • The authors contribute valuable genomic data from Ethiopia, where SARS-CoV-2 sequencing is underrepresented.

  • However, the study’s novelty should be better emphasized. The Delta variant’s global features are well known; thus, the unique local findings (e.g., mutation patterns distinct to Ethiopian samples, epidemiologic correlates, regional introductions) must be highlighted.

  • Please discuss how these results differ from Delta data from neighboring African countries (Kenya, Sudan, Uganda) or from GISAID African clades.

2. Study Design and Sampling

  • The sampling framework (AFI surveillance network) is clear, but the selection of samples for sequencing (48 out of 101 PCR-positive; 26 final with Ct <30) is somewhat small.

  • Authors should include a flow diagram or table summarizing:

    • Total samples tested

    • SARS-CoV-2 positives

    • Number sequenced

    • Ct thresholds

    • Criteria for inclusion/exclusion

  • Clarify whether sequencing depth, coverage percentage, and quality control metrics were assessed (e.g., % genome recovered, average coverage >30×).

3. Methods Clarity

  • The “Methods and Material” section needs restructuring according to Genes guidelines:

    • Ethical approval can be shortened.

    • Include library preparation (kit name, platform: Illumina/Nanopore), read trimming, assembly, and variant calling pipelines (e.g., Nextclade, Pangolin, IQ-TREE for phylogeny).

    • Specify reference genome version (e.g., NC_045512.2) and software versions.

    • Mention quality control thresholds (number of Ns, ambiguous sites, mean read length).

    • Provide GenBank or GISAID accession numbers for sequences.

4. Results Section

  • Add a results table summarizing the main lineages, mutations, and sub-lineages detected.

  • The percentage reporting (AY.120 = 46.2%, B.1.617.2 = 26.9%) is good, but please provide absolute counts (n) for clarity.

  • The statement “The number of mutations increased from Alpha (~35) to Delta (~42) to Omicron (~56)” is interesting — provide details on which mutations were counted (whole genome or Spike only?).

5. Discussion

  • Expand the discussion on regional epidemiology:

    • How did these Delta sub-lineages coincide with the third wave in Ethiopia?

    • Any correlation with vaccination coverage, case severity, or travel routes?

  • Interpret the mutation findings biologically: e.g., L452R, T478K, and P681R → increased infectivity and immune evasion; E156-F157 deletion → antigenic drift.

  • Clarify how your findings align or differ from those in neighboring African genomic reports

6. 

Language and Structure

  • The manuscript is readable but has frequent grammatical and structural issues. Examples:

    • “The findings observed in this study is indicative…” → “The findings observed in this study are indicative…”

    • “...causing major COVID-19 outbreaks that put a pressure…” → “...causing major COVID-19 outbreaks that placed pressure…”

  • Consider professional English editing or MDPI’s language service.

  • Ensure all references are cited in order and formatted in MDPI (Genes) style: numbered [1–30], not (1) (2)(3).

Author Response

Comment 1. The manuscript presents genomic surveillance data of SARS-CoV-2 variants collected from Acute Febrile Illness (AFI) patients in Ethiopia between February 2021 and June 2022, focusing on the predominance and genomic characteristics of the Delta (B.1.617.2 and AY.120) lineages. The study is relevant and timely, as genomic data from sub-Saharan Africa remain limited, and understanding lineage diversity is important for pandemic preparedness and vaccine-response monitoring.

The paper is generally well organized, but it needs major revisions to improve scientific clarity, data presentation, and interpretation. The methodology, results, and discussion sections require additional detail to meet Genes’ expected rigor. Some phrasing and grammar also need refinement for scientific English.

Response 1. I appreciate for reviewing the article and provided such a valuable and relevant comments/suggestions. I made a revision on the article addressing each of your feedbacks.

Comment 2. Novelty and Scientific Value

  • The authors contribute valuable genomic data from Ethiopia, where SARS-CoV-2 sequencing is underrepresented.
  • However, the study’s novelty should be better emphasized. The Delta variant’s global features are well known; thus, the unique local findings (e.g., mutation patterns distinct to Ethiopian samples, epidemiologic correlates, regional introductions) must be highlighted.

Response 2.  I agreed with your suggestion to emphasize on the novelty and I made a correction by adding the following paragraph in the discussion section.

Predominance of the AY.120 Sub-lineage:

  • The most distinctive finding in this study was the predominance of the Delta sub-lineage AY.120, which accounted for 46.2% (n= 22) of all sequenced samples. This suggests that AY.120 was not just a sporadic import but likely underwent significant local transmission within Ethiopia, potentially becoming a locally adapted strain during the study period. Page 10, Line number 254-258

Distinct Mutation Profile:

  • The Delta variants detected (including AY.120 and B.1.617.2) carried the characteristic suite of spike protein mutations such as L452R, T478K, P681R, and the double deletion E156-F157-, which are linked to increased transmissibility and immune evasion. While these are hallmarks of the Delta variant globally, their consistent presence in the Ethiopian clusters confirms the circulation of highly infectious strains and provides a baseline of the specific mutations circulating in the country. Page 11, LINE Number: 307-312

Evidence of Multiple Introductions and Local Transmission:

  • The phylogenetic analysis indicates that the Delta variants in Ethiopia did not originate from a single introduction event. Instead, the data show clustering patterns suggestive of multiple independent introductions from international sources, followed by established local transmission chains. This underscores the role of global travel in virus importation and the subsequent community spread within Ethiopia. Page 11, Line Number: 321-330.

Comment 3. Please discuss how these results differ from Delta data from neighboring African countries (Kenya, Sudan, Uganda) or from GISAID African clades.

Response 3. I agreed with your suggestion and included in the discussion part as follows:

  • Regionally, Ethiopia's experience aligns with broader African trends where Delta dominated mid-2021. However, the specific dominance of the AY.120 sub-lineage in Ethiopia differs from genomic reports in neighboring countrieslike Kenya and Uganda, which reported a more diverse mix of Delta sub-lineages (e.g., AY.36, AY.46). This unique sub-lineage profile suggests that Ethiopia's epidemic was shaped by specific international introduction events, potentially from Europe or Asia where AY.120 was also detected, followed by localized transmission rather than cross-border spread from immediate neighbors. This underscores the importance of local genomic surveillance to track distinct transmission dynamics that global or regional data may not capture. Page 10, Line number 258-266.

Comment 4. Study Design and Sampling

  • The sampling framework (AFI surveillance network) is clear, but the selection of samples for sequencing (48 out of 101 PCR-positive; 26 final with Ct <30) is somewhat small.

Response 4. I agree with this, its highlighted as a limitation in discussion part of the manuscript. Page 11, Line number 339-342

Comment 5. Authors should include a flow diagram or table summarizing:

    • Total samples tested
    • SARS-CoV-2 positives
    • Number sequenced
    • Ct thresholds
    • Criteria for inclusion/exclusion

Response 5. I agree and included the flow diagram below. Page 3, Line number 92-94

  •  

Comment 6. Clarify whether sequencing depthcoverage percentage, and quality control metrics were assessed (e.g., % genome recovered, average coverage >30×).

Response 6. The methods and results sections provide specific data on these metrics:

      • "The results showed that the 26 isolates had high sequencing depth, with an average depth of coverage ranging from 279 to 1014... The average genome coverage was 99.4%, confirming that the data were suitable for downstream genomic analysis." (Page 5, Lines 135-140)
        "Genome coverage of consensus genomes > 95% was used for the downstream analysis." (Page 4, Lines 102-103)
    • Quality Control Metrics:
    • "The nucleotide identity and amino acid identity were consistently high across all isolates, with an average of 99.8% and 99.7%, respectively." (Page 5, Lines 138-139)
      "Samples with Ct values of 30 or less were deemed uniformly suggestive of a substantial viral load for the purposes of genome sequencing..." (Page 3, Lines 81-83)
      "Consensus genome that contains less than 3000 missing or ambiguous (N’s) nucleotide was used for variant analysis." (Page 4, Lines 111-112)
  • I made a modification in the text as follows:
    • The study employed rigorous and well-documented quality control measures. An average depth of coverage ranging from 279 to 1014, with excellent genome coverage (>95%), and high nucleotide identity (>99.8%) collectively confirm that the genomic data generated was of high quality and that the findings on mutations and variants are reliable. Page 5, line number 148-152
    •  

Comment 7. Methods Clarity

  • The “Methods and Material” section needs restructuring according to Genes guidelines:
    • Ethical approval can be shortened.

Response 7. I agreed and modified as follws:

      • Ethical clearance was obtained from Ethiopian Public Health Institute (EPHI-IRB-254-2020) and Addis Ababa University (IRB/19/025). All participants provided written informed consent prior to enrollment. Line number 63-65

Comment 8. Include library preparation (kit name, platform: Illumina/Nanopore), read trimmingassembly, and variant calling pipelines (e.g., Nextclade, Pangolin, IQ-TREE for phylogeny).

    • Specify reference genome version (e.g., NC_045512.2) and software versions.
    • Mention quality control thresholds (number of Ns, ambiguous sites, mean read length).
    • Provide GenBank or GISAID accession numbers for sequences.

Response 8. I agree with the suggestion given by you and restructured it as follows (Page 3: Line number 96-126)

Whole-Genome Sequencing and Bioinformatics Analysis

Whole-genome sequencing was performed on SARS-CoV-2 positive samples with a cycle threshold (Ct) value of less than 30. Viral RNA was converted to cDNA using the LunaScript RT SuperMix (New England Biolabs). Subsequently, the SARS-CoV-2 genome was amplified using the Midnight RT PCR Expansion Kit (EXP-MRT001, Oxford Nanopore Technologies), which generates approximately 1.2 kb overlapping amplicons. Library preparation was carried out using the Rapid Barcoding Kit 96 (SQK-RBK110.96, Oxford Nanopore Technologies), and sequencing was performed on the MiniON platform using R9.4.1 flow cells (FLO-MIN106). Real-time basecalling and demultiplexing were executed within the MinKNOW software (v. 23.04.13).

Raw FAST5 reads were basecalled and demultiplexed using Guppy (v. 6.4.6) within the MinKNOW interface. The resulting FASTQ files were processed using the wf-artic workflow within the Nextflow-based nf-core/viralrecon pipeline (v.2.6.0), which is optimized for amplicon-based SARS-CoV-2 data. This pipeline performed read trimming, alignment to the SARS-CoV-2 reference genome Wuhan-Hu-1 (NC_045512.2) using minimap2 (v.2.24), and consensus genome generation using ivar (v.1.3.1). Quality control thresholds for the consensus genomes included a minimum depth of coverage of 30x, a genome coverage of >95%, and fewer than 3,000 ambiguous bases (N's). The mean read length after trimming was approximately 1,200 bp, consistent with the expected amplicon size.  SARS-CoV-2 raw reads obtained from Oxford Nanopore sequencing machine were assembled by using Genome Detective Platform Version 2.15.1 (https://www.genomedetective.com/) to generate consensus genomes.

Variant Calling and Phylogenetic Analysis

    • Variant calling and lineage assignment were conducted using two primary tools. The consensus sequences were analyzed with Nextclade CLI (v.3.8.2) for quality control, clade assignment, and identification of amino acid substitutions and deletions. PANGOLIN (v.4.3, accessed via the web interface at pangolin.cog-uk.io) was used for dynamic Pango lineage classification. For phylogenetic analysis, a maximum-likelihood tree was inferred using the Augur pipeline (v.23.1), which employs IQ-TREE (v.2.2.0) with the General Time Reversible (GTR) model of nucleotide substitution. The resulting tree was visualized using Auspice.us. The complete genome sequences generated in this study have been deposited in the GISAID database and are publicly available.

Comment 9. Results Section

  • Add a results table summarizing the main lineages, mutations, and sub-lineages detected.

Response 9. I agreed and included the below summary table:

  • Table: Summary of SARS-CoV-2 Variants, Sub-lineages, and Characteristic Mutations Detected in Ethiopia (2021-2022)
  •  

WHO Variant (Clade)

Pango Lineage (Sub-lineage)

Frequency n (%) [N=26]

Characteristic High-Frequency Spike Protein Mutations (>75%)

Characteristic Deletions (>75%)

Alpha (201)

B.1.1.7

2 (7.7%)

N501Y, D614G, P681H

H69-, V70-, Y144-

Delta (211)

AY.120

12 (46.2%)

T19R, G142D, R158G, L452R, T478K, D614G, P681R, D950N

E156-, F157-

B.1.617.2

7 (26.9%)

T19R, G142D, R158G, L452R, T478K, D614G, P681R, D950N

E156-, F157-

AY.32

2 (7.7%)

T19R, G142D, R158G, L452R, T478K, D614G, P681R, D950N

E156-, F157-

Omicron (21K)

BA.1.1, BA.1.17, BA.1

3 (11.5%), One (3.8%) for each sub linage

A67V, T95I, G142D, Y145D, G339D, S371L, S373P, S375F, K417N, N440K, S477N, T478K, E484A, Q493R, Q498R, N501Y, Y505H, D614G, H655Y, N679K, P681H, N764K, D796Y, N856K, Q954H, N969K, L981F

E27-, N28-, A29-, H69-, V70-, G142-, V143-, Y144-, N211-

Comment 10. The percentage reporting (AY.120 = 46.2%, B.1.617.2 = 26.9%) is good, but please provide absolute counts (n) for clarity.

Response 10. I made a correction as follows (line number 160-162) and included it in table 5.

    • AY.120 (n=12, 46.2%) and B.1.617.2 (n=7, 26.9%) under the Pangolin Lineage naming systems. Omicron and Alpha (Pango B.1.1.7) variants accounted for 11.5% (n=3) and 7.7% (n=2)

Comment 11. The statement “The number of mutations increased from Alpha (~35) to Delta (~42) to Omicron (~56)” is interesting — provide details on which mutations were counted (whole genome or Spike only?).

Response 11. The statement "The number of mutations increases from Alpha (~35 avg) to Delta (~42 avg) to Omicron (~56 avg)" refers to the average number of mutations across the entire genome for each variant, relative to the original Wuhan-Hu-1 reference strain (NC_045512.2). Line number 221-223.

Comment 12. Discussion

  • Expand the discussion on regional epidemiology:
    • How did these Delta sub-lineages coincide with the third wave in Ethiopia?

Response 11. I agree with your suggestion and I made a modification as follows

  • The genomic data from this study provide a molecular snapshot of the COVID-19 pandemic in Ethiopia, confirming that the Delta variant was the primary driver of the country's third wave from July to October 2021. The overwhelming predominance of Delta (80.8%), particularly the 120 sub-lineage, suggests this sub-lineage may have had a selective advantage for local transmission, leading to a significant community outbreak during this period. Page 9-10, line number 242-247.

Comment 12. Any correlation with vaccination coverage, case severity, or travel routes?

Response 12. I agree with your suggestion and I made a modification as follows.

  • While the study does not perform a direct statistical correlation, the timing of the Delta wave coincided with a period of low vaccination coveragein Ethiopia, which likely facilitated the variant's rapid spread. The high proportion of inpatients (90%) among sequenced cases is consistent with global reports of increased clinical severity associated with the Delta variant, including a higher risk of hospitalization compared to previous variants. Page 10, Line number 285-290

Comment 13. Interpret the mutation findings biologically: e.g., L452R, T478K, and P681R → increased infectivity and immune evasion; E156-F157 deletion → antigenic drift.

Response 13. I agree with your suggestion and I made a modification as follows.

  • The mutation profile of the circulating Delta variants explains their success. The critical spike mutations L452Rand T478K are known to enhance binding to the human ACE2 receptor and confer significant immune evasion from neutralizing antibodies. The P681R mutation facilitates more efficient viral entry into cells by enhancing spike protein cleavage. Furthermore, the E156-F157 deletion is thought to contribute to antigenic drift by removing an epitope, thereby weakening the antibody response from prior infection or vaccination. Page 11, Line number 309-315.

Comment 14. Clarify how your findings align or differ from those in neighboring African genomic reports

Response 14. I agree with your suggestion and I made a modification as follows:

  • Regionally, Ethiopia's experience aligns with broader African trends where Delta dominated mid-2021. However, the specific dominance of the AY.120 sub-lineage in Ethiopia differs from genomic reports in neighboring countries like Kenya and Uganda, which reported a more diverse mix of Delta sub-lineages (e.g., AY.36, AY.46). This unique sub-lineage profile suggests that Ethiopia's epidemic was shaped by specific international introduction events, potentially from Europe or Asia where AY.120 was also detected, followed by localized transmission rather than cross-border spread from immediate neighbors. This underscores the importance of local genomic surveillance to track distinct transmission dynamics that global or regional data may not capture. Page 10, Line number 258-266.

Comment 15. Language and Structure

  • The manuscript is readable but has frequent grammatical and structural issues. Examples:
    • “The findings observed in this study is indicative…” → “The findings observed in this study are indicative…”
    • “...causing major COVID-19 outbreaks that put a pressure…” → “...causing major COVID-19 outbreaks that placed pressure…”.

Response 15. Thank you for such relevant inputs. I made a correction in all grammatical and structural issues.

  • Consider professional English editing or MDPI’s language service.

Comment 16. Ensure all references are cited in order and formatted in MDPI (Genes) style: numbered [1–30], not (1) (2)(3).

Response 16. Thank you for such relevant inputs. I made a correction in all references

Reviewer 2 Report

Comments and Suggestions for Authors

In this manuscript, the authors provide a sequencing-based analysis of SARS-CoV-2 isolates in of acute febrile illness (AFI) patients in four Ethiopian hospitals from a study period in February 2021-June 2022.  After selection criteria based upon Ct values less than 30, 26 samples were sequenced. While a larger sample size would be desired, the patient demographics cover a range of ages, gender balance, and symptoms. Genomic analysis revealed the Delta variant was predominant in Ethiopia, accounting for over 80% of sequenced samples from AFI patients, with AY.120 being the most common sub-lineage. The virus showed a clear evolutionary trajectory, with the number of mutations increasing from Alpha to Delta to Omicron, and phylogenetic evidence suggested multiple introductions and local transmission of the virus. Overall, the methodology is standard, and the findings are presented well.  Though additional samples would strengthen the ability to do enhanced statistics, this is an interesting study to map SARS-CoV-2 infection in Ethiopia. I have only a few comments:

Line 174: the finding of Omicron in October 2021 is earlier than what is regarded as the earliest confirmed sample in South Africa on November 8, 2021.  This could be an important finding if the sampling date is verified and may warrant additional mention in the text.

While the authors state they are willing to provide the sequencing data, I would encourage them to go ahead and make them available in an online repository.

Minor text edits:
Change the citation format to have everything in one set of parentheses and abbreviated with 3 or more contiguous, eg. line 37 should be (1-3)
Line 29: change to "are indicative"
Line 188: capitalize "table 4"
Lines 202-205: the format of this paragraph should be changed for consistency with the rest of the main text
Line 211: change to "genomes were characterized"
Lines 251, 274: remove the double open parentheses
Line 256: place a space before the parentheses
Line 325: change "scares" to "scarce"

Author Response

Comment 1.

  • In this manuscript, the authors provide a sequencing-based analysis of SARS-CoV-2 isolates in of acute febrile illness (AFI) patients in four Ethiopian hospitals from a study period in February 2021-June 2022.  After selection criteria based upon Ct values less than 30, 26 samples were sequenced. While a larger sample size would be desired, the patient demographics cover a range of ages, gender balance, and symptoms. Genomic analysis revealed the Delta variant was predominant in Ethiopia, accounting for over 80% of sequenced samples from AFI patients, with AY.120 being the most common sub-lineage. The virus showed a clear evolutionary trajectory, with the number of mutations increasing from Alpha to Delta to Omicron, and phylogenetic evidence suggested multiple introductions and local transmission of the virus. Overall, the methodology is standard, and the findings are presented well.  Though additional samples would strengthen the ability to do enhanced statistics, this is an interesting study to map SARS-CoV-2 infection in Ethiopia. I have only a few comments:

Response 1. I appreciate for reviewing the article and provided such a valuable and relevant comments and suggestions. I made a revision on the article addressing each of you feedbacks.

Comment 2. Line 174: the finding of Omicron in October 2021 is earlier than what is regarded as the earliest confirmed sample in South Africa on November 8, 2021.  This could be an important finding if the sampling date is verified and may warrant additional mention in the text.

Response 2. I thank you so much in identifying it, appreciated.  I crosschecked my epi-data and I found it was by mistake; it was detected in November 2021 and made a correction in figure 2, and in the text Line 171. I will attach both the epi data and sequence data as a supplementary file.

Comment 3. While the authors state they are willing to provide the sequencing data, I would encourage them to go ahead and make them available in an online repository.

Response 3. I’m working on it and waiting for activation of my account on GISAID; it may take few more days to activate it as per the email I received from GISAID.

Comment 4. Minor text edits:
Change the citation format to have everything in one set of parentheses and abbreviated with 3 or more contiguous, eg. line 37 should be (1-3).

Response 4. I made a correction accordingly.

Comment 5. Line 29: change to "are indicative"

Response 5. I made a correction accordingly.

Comment 6. Line 188: capitalize "table 4"

Response 6. I made a correction accordingly.

Comment 7. Lines 202-205: the format of this paragraph should be changed for consistency with the rest of the main text.

Response 7. I made a correction accordingly.

Comment 8. Line 211: change to "genomes were characterized"

Response 8. I made a correction accordingly.

Comment 9. Lines 251, 274: remove the double open parentheses

Response 9. I made a correction accordingly.

Comment 10. Line 256: place a space before the parentheses

Response 10. I made a correction accordingly.

Comment 11. Line 325: change "scares" to "scarce"

Response 11. I made a correction accordingly.

Round 2

Reviewer 1 Report

Comments and Suggestions for Authors

The authors have successfully covered the reviewer's suggestions.

Author Response

Comment 1. In figure 1, the text is small and unclear in terms of visualization. 
Response 1. I thank you for your feedback. I made a correction accordingly. Figure 1.

Figure 1. Flow Diagram of Sample Processing to Genomic Analysis

Comment 2. In figure 2, check the format of months in the x-axis (does it have strange symbols?). 
Also in the legend, you may want to indicate "Variant of Concern" instead of just "Variant" (there are many variants, not only VOCs). 
Response 2. I thank you for your feedback. I made a correction a accordingly. Figure 2. 

Figure 2. Temporal Distribution of SARS-CoV-2 WHO variant of concern (alpha, Delta and Omicron) by month during in Ethiopia (2021-2022).

Comment 3. In figure 3, what is the meaning of "clade_who"?, maybe use again "Variant of Concern". 
Clarify the meaning of "Divergence" in the x-axis, is it evolutionary or real time?
Response 3. I made a correction and clarified it in the legend as follows:

Figure 3. Phylogenetic tree visualized using Auspice.us and showing the evolutionary relationships of SARS-CoV-2 variant of concern. The tree is rooted on the reference genome NC_045512.2 (Wuhan-Hu-1). Major clades are labeled using the World Health Organization (WHO) nomenclature system (clade_who), which designates Variants of Concern (VOCs) Alpha, Delta, and Omicron. The horizontal axis (Divergence) indicates the genetic distance from the root, measured in nucleotide substitutions per site. Sample names are shown for sequences within the defined clades.
Comment 4. I think the Discussion section could be improved by comparing the detected mutations with those detected in previous studies around the world and already associated to the different VOCs, for example see the review article https://www.mdpi.com/2073-4425/14/2/407.  
Response 4. I think all the major finding were discussed in comparison to the available regional and global studies, publications. I am afraid it will be bulky if I’m going to add more. The regional and global comparison presented in the discussion part; Line number 289-297; Line number 298-307.
Comment 5. The Introduction section is too short, it could be increased, providing more information about expansion and evolution of the virus (the previously indicated review could also help for this), as well as the current situation in Ethiopia.
Response 5. I agree with your suggestion and modified the introduction section, included the situation in Ethiopia as well as providing more information about expansion and evolution of the virus. Line number 37-39, Line number 45-57, and Line number 64-73
